# AI-POWERED ASSESSMENT FRAMEWORK FOR SKILL-ORIENTED ENGINEERING LAB EDUCATION

## ABSTRACT

Practical lab education in computer science often faces challenges like plagiarism, lack of proper lab records,unstructured lab conduction,inadequate execution and assessment,lack of practical learning,limited student engagement, and absence of progress tracking for both students and faculties causing graduates with insufficient hands-on skills. In this paper, we introduce **AsseslyAI**, it tackles these challenges through online lab allocation, unique lab problem for each student, integrates AI-proctored viva evaluations, and gamified simulators to enhance engagement and conceptual mastery. While existing platform are generating questions on the basis of topics, our framework fine-tunes on a 10k+ Question-Answer dataset built from AIML lab questions to dynamically generate diverse, code-rich assessments. Validation metrics show high Question-Answer similarity, ensuring accurate answers and non-repetitive questions. By unifying dataset-driven question generation, adaptive difficulty, plagiarism resistance, and evaluation in a single pipeline, our framework advances beyond traditional automated grading tools and offers scalable path to produce genuinely skilled graduates.

## 1 INTRODUCTION

Practical laboratories are the foundation of computer science and engineering education. While theoretical knowledge can be gained through lectures and written examinations, laboratory exercises focus on developing student problem solving as well as programming skills, and applied understanding of core concepts. However, in many institutions, students often finish their courses without actively engaging in laboratory work. Cheating from other classmates and online platforms, lack of proper evaluation method make this issue bigger. This results in graduates with degrees that lacks practical knowledge required by industries. Recently efforts have been made to modernize assessment by introducing automated grading systems, online coding platforms, and plagiarism detection tools. Though these methods address part of the problem, they do not ensure authentic skill development. The existing systems struggles in personalizing the laboratory work, preventing collusion, and assessing students' concepts beyond coding output. Their is a need of an intelligent framework that can increase genuine learning, reduce academic unfairness and give insights of students performance to the faculty. In this work, we propose a solution to this problem an AI based Computer Science Laboratory management system. It generates unique laboratory questions for each student, keeps track of their progress, and also conducts a viva based on the question. Working, Faculties will insert keywords such as the 'Supervised learning' or 'deep neural networks' for which the system dynamically generates unique questions for each student. This ensures that no two students get identical questions, which will reduce the opportunity of copying from others, and the student will try to do it independently. For strengthening this system more, we have introduced an AI-Viva assistant that conducts a viva voice on the assigned question, evaluating not only the coding outcomes but also the conceptual understanding of the topic and reasoning of the students. Our contributions are threefold:

- Introduced a framework for lab that generates unique questions on the bases of faculty-defined input keyword and difficulty level.

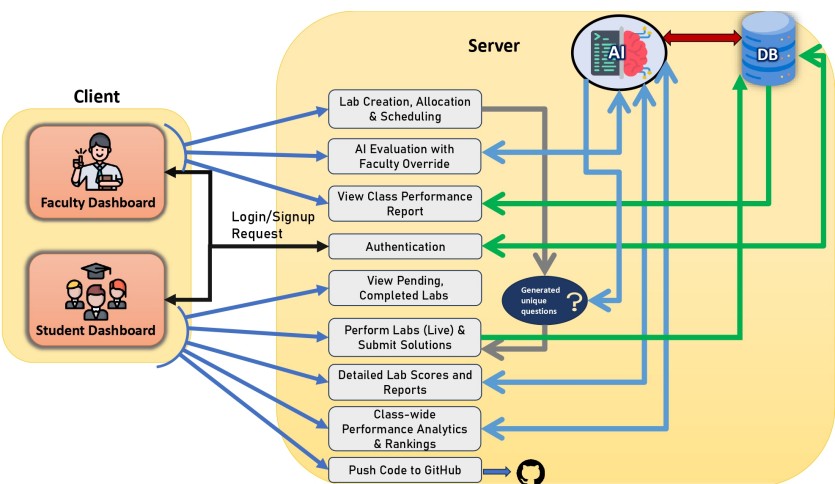

Figure 1: Over View of Ai-Lab Management Systems.

- The Faculty can choose between proctored and non-proctored modes. In both modes, students are assessed holistically beginning with basic conceptual checks through gamified quizzes, followed by coding tasks, and concluding with an interactive viva that evaluates their depth of understanding and reasoning.

- Through this system the enduring challenge of unskilled graduates by improving engagement, discouraging cheating, and providing richer feedback on student performance is addressed.

After combining technology with teaching, our system provides students a structured way to improve their practical skills in computer science. By adding anti-cheating features and viva tests, our system fixes the gap of normal grading system and helps students to become more serious about their labs.

## 2 RELATED WORK

### 2.1 LARGE LANGUAGE MODELS AND EDUCATIONAL DATASETS

Recent advances in large language models (LLMs) have transformed programming education by enabling automated code generation, grading, and tutoring. Models such as Codex (Chen et al., 2021), CodeT5 (Wang et al., 2021), and and Code Llama (Roziere et al., 2023), built on Transformer architecture (Vaswani et al., 2017), demonstrate strong capabilities in program synthesis and explanation. In educational contexts, Jukiewicz (2024) showed that ChatGPT could grade programming assignments with consistency comparable to human instructors, providing advantages in efficiency and personalization. However, risks of hallucinations and misinterpretations remain, underscoring the need for domain-specific fine-tuning and human oversight.

Datasets play a central role in enabling these advances. Large-scale resources such as CodeNet (Puri et al., 2021) provide 14 million programs across multiple languages, while repositories from Codeforces, LeetCode, and AtCoder are commonly mined for AI-driven research. However, these datasets are designed primarily for competitive programming, not for alignment with pedagogical goals. This motivates the need for curated, difficulty-labeled datasets that support education-specific objectives such as the 10,000-question dataset we construct and use for fine-tuning in our work.

### 2.2 AUTOMATIC QUESTION GENERATION AND DIFFICULTY CALIBRATION

Automatic question generation (QG) in programming education remains comparatively underexplored compared to grading. General-purpose QG approaches have relied on templates, rule-based natural language processing, or neural text generation methods (Kurdi et al., 2020). In computer science education, researchers have investigated program synthesis approaches to generate exercises (Pan et al., 2019) and metrics such as cyclomatic complexity, variable count, and nesting depth to assign

difficulty (Kumar & Singh, 2019; Moreno et al., 2012). Pedagogical frameworks also leverage Bloom's taxonomy to map exercises to cognitive levels (Anderson & Krathwohl, 2001).

Despite these efforts, most existing QG approaches either depend on static templates or produce limited problem variations, making them prone to plagiarism and insufficiently adaptive to individual learners. Difficulty calibration remains a major challenge, as many systems lack fine-grained control over question difficulty or alignment with faculty-driven learning outcomes.

### 2.3 Personalization, Plagiarism Prevention, and Adaptive Learning

A long-standing goal in education is personalization, with the aim of adapting exercises to the needs of individual learners. Adaptive learning platforms (Le & Pinkwart, 2015) attempt to dynamically adjust question difficulty based on student performance, while plagiarism detection systems such as MOSS (Schleimer et al., 2003) and JPlag (Joy & Luck, 1999) help ensure academic integrity.

However, existing systems typically rely on fixed problem banks and randomized instances, which cannot guarantee unique keyword-driven exercises for each learner. This limitation reduces personalization, increases opportunities for copying in large classes, and restricts the ability of faculty to tailor lab problems around specific concepts or skills. A system capable of generating unique, faculty-driven exercises at scale would directly address these shortcomings.

### 2.4 Automated Assessment and Feedback in Programming Education

Automated grading approaches are generally classified into four categories: static, dynamic, hybrid, and AI-driven methods. Static analysis checks code properties like style, plagiarism, and syntax accuracy. Dynamic analysis executes programs against instructor-defined test cases, ensuring functional correctness but often neglecting efficiency or conceptual mastery, as seen in platforms like HackerRank or Codeforces. Hybrid methods integrate both signals for more robust evaluation (Ihantola et al., 2010), while AI-based systems exploit clustering and error-pattern mining to provide richer diagnostics Zhi et al. (2019).

Feedback is consistently highlighted as central to effective assessment. Early automated systems typically provided only binary pass/fail outcomes, which limited their pedagogical value (Shute, 2008). Recent frameworks aim to deliver formative, constructive feedback. For example, the Pythia platform (Singh et al., 2019) integrates automated grading with personalized hints, helping students move beyond syntax errors toward conceptual understanding. Nevertheless, most automated assessment systems still emphasize correctness, with limited support for higher-order skills, creativity, or conceptual reasoning.

## 3 Dataset

### 3.1 Dataset Construction

We constructed a synthetic dataset of 10,000 programming questions targeting core topics in machine learning (ML) and artificial intelligence (AI). The question set of the synthetic dataset was created using a hybrid approach, a combination of Python scripts and ChatGPT (OpenAI, 2023) to ensure diversity in question phrasing and coverage. The corresponding answers of the synthetic dataset were generated in batches of 1,000 using Perplexity AI (Srinivas et al., 2023), followed by manual review for accuracy and consistency.

The synthetic dataset covers both classical Machine Learning algorithms such as Decision Trees, Random Forests, k-Nearest Neighbors, Support Vector Machines, Regression and Deep Learning architectures like Convolutional Neural Networks, Recurrent Neural Networks, Long Short-Term Memory networks, feedforward networks, and optimization techniques.

Each record in the synthetic dataset contains six fields:

(i) **Id:** A unique identifier assigned to each question.
(ii) **question:** The generated programming lab question text.
(iii) **answer:** The generated lab answers in Python programming language.

(iv) **category:** Difficulty label (*Easy*, *Medium*, or *Hard*) assigned by faculty annotators.
(v) **marksAI:** Automated score (0-100) generated by our AI-based evaluation model.
(vi) **marksFaculty:** Human-assigned score (0-100) provided by a faculty member.

# 4 LAB MANAGEMENT SYSTEMS

## 4.1 CLIENT INTERFACES

In this system design, we provide two user roles: Faculty and Student. Each user group has dashboard with respective feature and responsibility according to their role.

### 4.1.1 FACULTY DASHBOARD

The faculty dashboard provides a simple and clear interface for faculty to create and manage laboratory tasks. The Faculty dashboard has a feature to create lab in which faculty has to provide information-topic keyword, difficulty level, viva duration, Lab mode, Lab description, and Lab instruction. Once created, laboratory work can be assigned to specific classes. A key feature we added is the ability to override or adjust AI-generated evaluations, ensuring that instructors retain Teacher authority. The dashboard also has a tab to Manage Allocations where faculty can remove or delete assigned laboratory tasks. The dashboard also provides access to class-wise performance analytics and comparative reports, enabling faculty to monitor learning outcomes across sections.

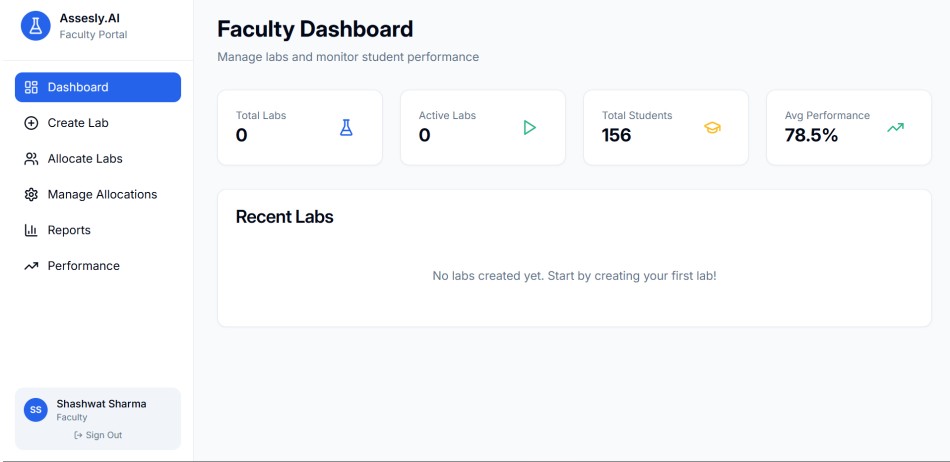

Figure 2: UI of Faculty Dashboard.

### 4.1.2 STUDENT DASHBOARD

The student dashboard is the primary interface for students to interact with assigned laboratories. Student Dashboard has an option of "My labs", as you can see in Figure-3, in My Labs the student can find all the assigned labs. After starting a laboratory task, each student is presented with a unique generated question aligned to the specified faculty keywords and difficulty level, ensuring that no two students receive identical problems. Students will write and submit code in supported programming languages and participate in AI-proctored viva test on the basis of the question assigned. After submission of code and viva, students receive detailed reports that include performance scores, conceptual feedback, and recommendations for improvement. For improving student's profile, integrating an external connector GitHub/Hugging face where students can upload their laboratory work.

## 4.2 SERVER COMPONENTS

The server-side architecture of Assesly.AI integrates multiple modules that ensure secure access, seamless lab administration, robust assessment of student submissions, and continuous performance

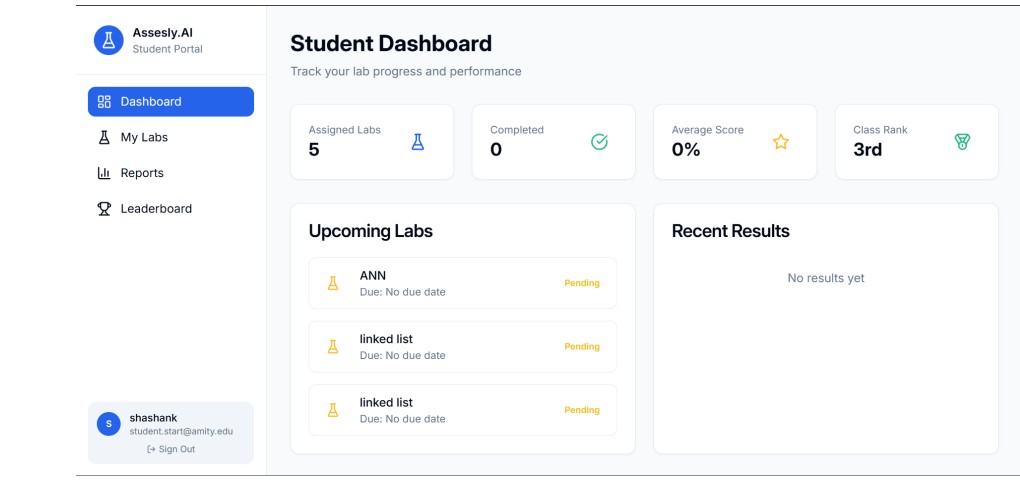

Figure 3: UI of Student Dashboard.

monitoring. These components coordinate to support automated grading, lab scheduling and allocation, progress analytics, and authentication.

### 4.2.1 AUTHENTICATION AND ACCESS CONTROL

The server ensures secure role-based access only through credential-based login, distinguishing faculty and student privileges. Faculty can create and allocate labs, monitor analytics, and override automated assessment scores, while students are limited to submissions, viewing results, and accessing feedback reports. Access control policies are enforced at both the interface and the API levels to ensure data security.

### 4.2.2 LAB MANAGEMENT AND EVALUATION ENGINE

This module handles the complete lab life-cycle, maintaining metadata such as lab identifiers, topics, deadlines, and allocations. Faculty can schedule labs, monitor active sessions in real time, and view completion status. Students are presented with pending, active, and completed tasks. Submissions are automatically evaluated using a fine-tuned XGBoost regressor(Chen & Guestrin, 2016). Detailed reports are generated for students (scores, quality feedback, errors, readability feedback) and faculty (aggregate performance trends, plagiarism alerts, and mastery insights). Faculty retain the ability to override automated scores where necessary.

### 4.2.3 PROGRESS PROFILING AND ANALYTICS

To support continuous learning and teaching improvement, the server maintains dynamic progress profiles for both students and faculty. For students, each lab submission updates their longitudinal performance graph, heatmaps, and cumulative completion statistics. For faculty, the progress profile tracks the count of labs conducted for different sections, measures consistency through heatmap, and visualizes student learning gains as an indicator of teaching effectiveness. These analyses are continuously updated after completion of every lab, allowing both student and faculty to monitor their progress accurately to identify gaps and make data-driven decisions.

### 4.3 AI QUESTION GENERATION PIPELINE, VIVA AND EVALUATION

The question generation pipeline transforms faculty inputs into personalized lab problems, ensuring that students receive assignments aligned with course objectives while maintaining individual uniqueness. By leveraging large language models fine-tuned for educational contexts, the pipeline supports scalable, adaptive, and pedagogically consistent lab design. It goes beyond static question banks by dynamically creating diverse, context-aware problems, helping reduce repetition and ensuring that every student faces a fair but distinct challenge.

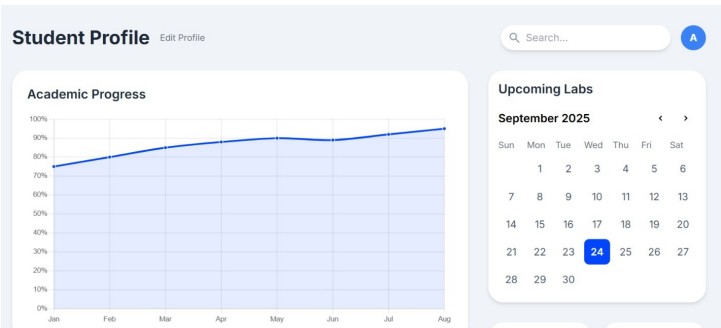

Figure 4: student academic progress

### 4.3.1 FACULTY-GUIDED QUESTION GENERATION

Faculty will provide topic keywords (e.g. : deep neural network, artificial neural network, etc,..) and also difficulty level (e.g.: easy, medium or hard), these input then are used for generation of question. Our system uses Codellama model which is fine-tuned on the dataset of 10,000 difficulty-labeled questions containing some AI topics, making sure that the question provided to the students are relevant and Lab based. Additionally, model is generating unique questions for each student in a class that will reduce cheating from other students.

### 4.3.2 AI BASED VIVA QUESTIONS

This system also have a viva test after submission of answer of the question. It integrates Ai for viva voice, AI will ask some questions based on the problem allocated to the student. This will help student in conceptual understanding and gradually student will become more confident for their interviews and external viva. Student responses, captured via automatic speech recognition (ASR), are scored against rubric-based answers to assess reasoning and domain knowledge. It also generates detailed analytics: students receive personalized feedback with scores and improvement suggestions, while faculty dashboards provide performance insights, plagiarism alerts, and rankings. Additionally, students can publish selected work to platforms like GitHub or Hugging face to showcase their skills.

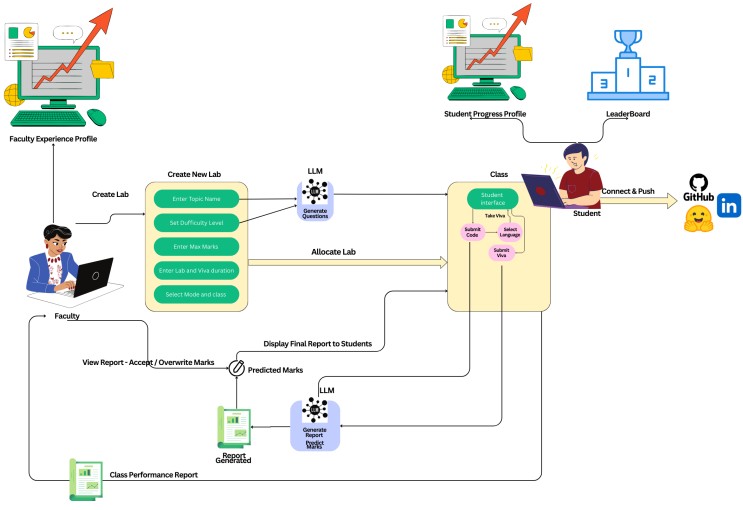

Figure 5: Detailed working Architecture of Ai-Lab Management Systems.

## 5 EXPERIMENTS

### 5.1 DATASET EVALUATION

The curated dataset fine-tunes our generator to produce seed questions from faculty-specified keywords and difficulty levels, and then generates unique, level-consistent variations for each student Le & Pinkwart (2015). This approach reduces redundancy and minimizes opportunities for collusion.

Unlike competitive programming datasets such as CodeNet Puri et al. (2021), our synthetic dataset is explicitly aligned with computer science lab curricula. It emphasizes implementation skills, conceptual understanding, and explainability-oriented evaluation rather than correctness alone, better reflecting the learning objectives of practical coursework.

We evaluated the reliability and quality of the synthetic 10,000-question dataset through multiple validation steps. Inter-annotator agreement (IAA) was measured between our automated scoring system (marksAI) and faculty-assigned scores (marksFaculty). Cohen's $\kappa$ for faculty annotations was positive, indicating substantial agreement, while comparison between marksAI and marksFaculty showed strong consistency, demonstrating that the automated assessment aligns well with human judgment.

We performed question-answer (QA) similarity analysis to further validate the synthetic dataset, which measured the semantic relevance between each generated question and its corresponding answer. The similarity score was high, confirming that the corresponding generated answers were strongly aligned with their questions. It ensures both relevance and correctness in the synthetic dataset.

### 5.2 FINE-TUNING THE QUESTION GENERATION MODEL

TThe question generation module was fine-tuned on a curated dataset using a transformer-based language model (e.g., Code Llama (Roziere et al., 2023)). Faculty-provided keywords and difficulty levels were used as conditional inputs to guide question synthesis. After generation, a semantic filtering pipeline was applied to remove near-duplicate questions and enforce diversity across students. Pairwise similarity analysis confirmed a substantial reduction in redundancy, resulting in unique and diverse question sets tailored for each learner.

### 5.3 FINE-TUNING THE ANSWER ASSESSMENT MODEL

For automated grading, we fine-tuned a code-understanding model using faculty-annotated student submissions, following practices similar to prior work on code evaluation models (Wang et al., 2023; Chen et al., 2021).. The model was trained to predict scores across multiple dimensions, including functional correctness, readability, and complexity, and to aggregate them into a final weighted grade. Evaluation was conducted through cross-validation, showing strong agreement with faculty grading and unbiased prediction patterns. Error analysis revealed that most deviations between predicted and faculty scores were minor, with rare larger deviations occurring in highly variable, sequence-heavy lab tasks (e.g., RNNs, Transformers, and RL). Faculty review of the model's explanations indicated they were pedagogically sound and actionable for student feedback.

## 6 RESULTS

### 6.1 RESULT OF DATASET VALIDATION

#### 6.1.1 QUESTION APPROPRIATENESS AND DIVERSITY

We first evaluate the quality and diversity of generated lab questions. Figure 6 shows the distribution of similarity scores in 10,000 pairs of generated question-answer (QA) questions. The majority of scores fall between 0.25 and 0.45, indicating moderate semantic overlap with high variance. This demonstrates that the model produces questions that are both aligned with faculty-provided keywords and sufficiently diverse across students. In contrast, template-based generators concentrated heavily around similarity scores $> 0.6$, reflecting limited variation. Faculty ratings further confirmed alignment, with an average appropriateness score of 4.7/5.

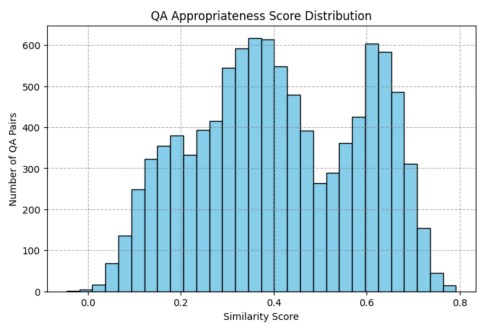

Figure 6: Question–Answer appropriateness score.

### 6.1.2 Assessment Accuracy

We evaluated the alignment between AI-based grading and instructor annotations across student submissions. Our approach demonstrated strong consistency. Figure 7 illustrates the results. The agreement analysis shows a high Pearson correlation of 0.914, indicating strong linear consistency between AI and faculty grading. The Spearman correlation of 0.892 further confirms strong rank-order agreement. Cohen's Kappa coefficient indicates substantial agreement ($\kappa = 0.69$).

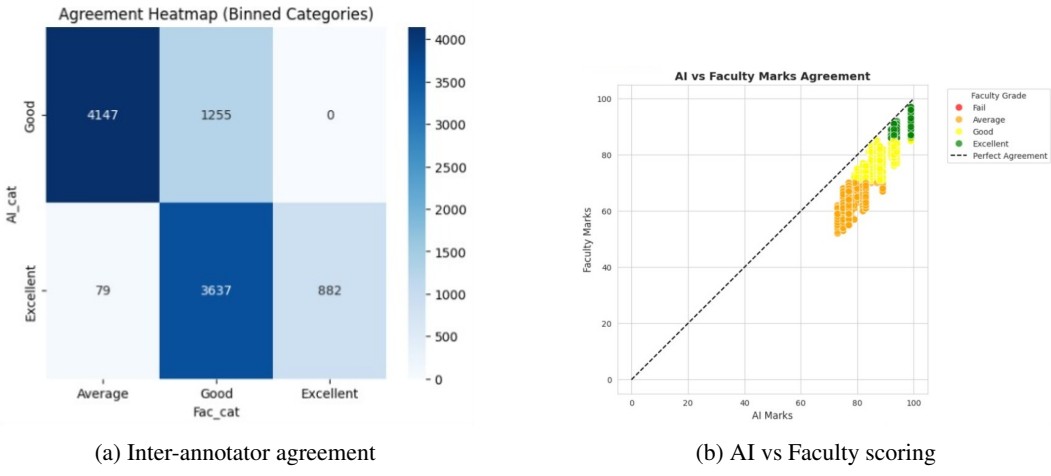

(a) Inter-annotator agreement                    (b) AI vs Faculty scoring

Figure 7: Assessment accuracy analysis: (a) consistency between human annotators, (b) alignment between AI and faculty marks.

### 6.1.3 Alignment Between AI and Faculty Scoring

As shown in Figure 7b, AI-assigned marks closely track faculty scores across submissions. The clustering of points along the diagonal confirms high agreement, particularly for higher-scoring submissions. Minor deviations appear in the mid-range (60–80), where borderline cases are more frequent. These patterns are consistent with our quantitative results (Pearson et al $r = 0.914$, Spearman et al $\rho = 0.892$), underscoring the reliability of our hybrid evaluation method in approximating instructor judgments.

### 6.2 Result of Finetuning

### 6.2.1 Evaluation Metrics

We evaluate the fine-tuned evaluator using cross-validation. The overall **RMSE = 3.37**, indicating that predictions differ on average by only ~3 marks on a 0–100 grading scale. The **$R^2$ = 0.861** shows that

the model explains approximately 86% of the variance in faculty grading, reflecting strong alignment (see Figure 8).

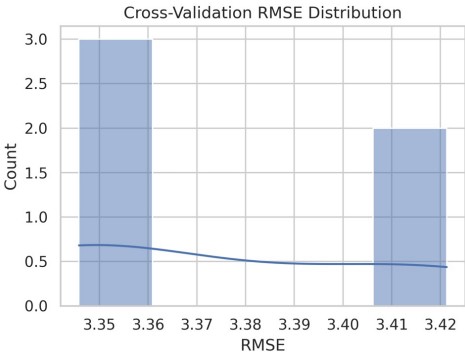

Figure 8: Cross-validation RMSE distribution across folds.

### 6.2.2 ERROR AND DISTRIBUTION ANALYSIS

The prediction error distribution (Figure 9a) follows a near-normal shape centered at zero, demonstrating unbiased grading. Most deviations fall within $\pm 5$ marks, with rare outliers extending to $\pm 10$. The predicted vs actual plot (Figure 9b) confirms strong alignment with faculty marks.

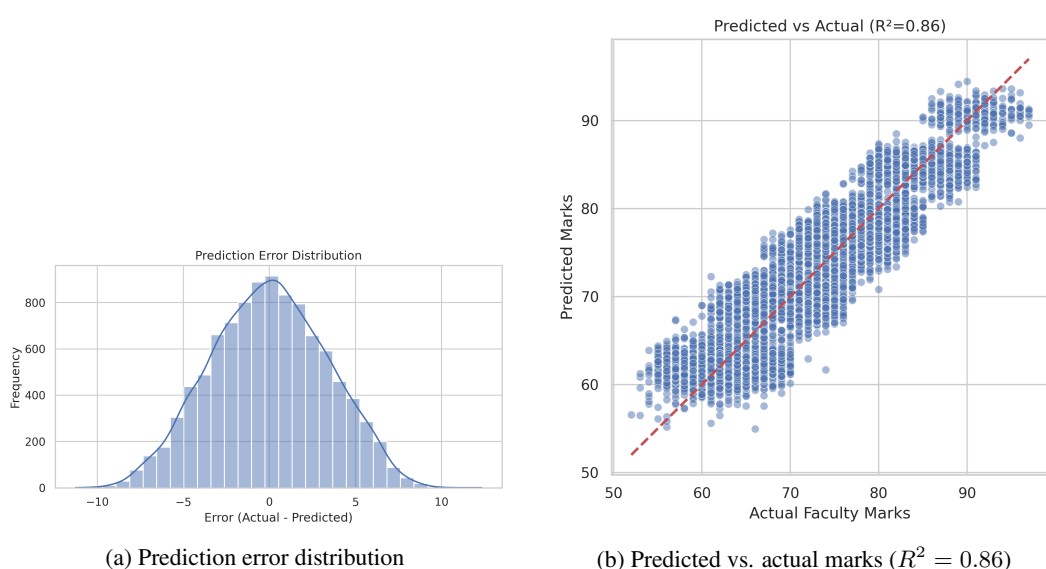

(a) Prediction error distribution      (b) Predicted vs. actual marks ($R^2 = 0.86$)

Figure 9: Error and distribution analysis of fine-tuned evaluator.

### 6.2.3 CASE STUDY OF WORST ERRORS

The top-10 largest deviations range between $\pm 9$–12 marks and are concentrated in *sequence-heavy labs* such as LSTM, GRU, Transformer, and Reinforcement Learning tasks. These labs are inherently variable in implementation, making exact alignment with human grading harder. For classical ML and NLP labs, however, the evaluator consistently aligns with faculty scores.

## REPRODUCIBILITY CHECKLIST

### 1. DATASET

(a) Source: Synthetic dataset of ∼10,000 programming lab questions (ML/AI topics). (b) Split: 90/10 for generator; 80/20 for evaluator (with 5-fold CV). (c) Size: ∼8,093 unique questions after deduplication, with answers. (d) Availability: Anonymous repository (to be released upon acceptance).

### 2. HYPERPARAMETERS

**Generator (CodeLlama-7B-Instruct + LoRA)** - Epochs: 1 (sanity) → 3 (full) - Batch size: 1 per device - Gradient accumulation: 8 - Learning rate: 1e-4 (AdamW) - Precision: FP16, 4-bit quantization (bitsandbytes) - LoRA: r=8, $\alpha$=32, dropout=0.05 - Max seq length: 512 (extended to 1024 for full runs) - Checkpointing: every 500 steps (keep last 3)

**Evaluator (XGBoost on SBERT embeddings)** - Embedding model: all-MiniLM-L6-v2 (384-dim) - n_estimators=500, max_depth=6 - learning_rate=0.05 - subsample=0.8, colsample_bytree=0.8 - tree_method=gpu_hist (or hist on CPU) - predictor=gpu_predictor - random_seed=42

### 3. MODELS

(a) Generator: LoRA fine-tuned CodeLlama-7B-Instruct. (b) Evaluator: Sentence-BERT embeddings + XGBoost regression. (c) Both models trained from scratch on our dataset (with PEFT for generator).

### 4. CODE AND DATA

(a) Code + dataset will be released at camera-ready in anonymized repository. (b) Scripts: preprocessing (deduplication, JSONL formatting), fine-tuning, evaluation, visualization. (c) Checkpoints + embeddings provided.

### 5. COMPUTE RESOURCES

(a) GPUs: NVIDIA A100 (training), RTX 3090 (experiments). (b) Preprocessing: ∼6 hours (10k records). (c) Generator fine-tuning: ∼12 hours (3 epochs). (d) Evaluator training: ∼1.5 hours per fold. (e) Inference: 1–2s per question (generator); <1s per evaluation (evaluator).

### 6. RANDOM SEEDS

(a) All experiments used fixed random seed = 42. (b) Cross-validation results are averaged across 5 folds.

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
