# OpenReview forum: "AsseslyAI: AI–Powered Assessment Framework for Skill-Oriented Engineering Lab Education"
_ICLR.cc/2026/Conference — ICLR 2026 Conference Desk Rejected Submission_

### Official Review · Reviewer_PVGm · 2025-10-27

**Soundness:** 2
**Presentation:** 1
**Contribution:** 1
**Rating:** 2
**Confidence:** 4

**Summary:**

This work focuses on building a framework for automatic assessment and feedback generation for students. Specifically, the authors built a dashboard for students and teachers to view the progress of students. They synthesized a set of 10K problems to avoid potential cheating issues as well.

**Strengths:**

+ The system if implemented correctly, should make meaningful contributions to the domain of intelligent tutoring systems.

**Weaknesses:**

- The framing of the work does not make sense -- what exactly is the motivation? It's just not coherent in the motivation framing of the paper. The lack of engagement in labs may not be directly associated with cheating -- it's another issue. In addition, why would these new assessment tools help address the issues mentioned before? It's just not clear in the introduction.
- Multiple validation processes or labeling processes require further details. For example, the difficulty levels are manually labeled -- it's a large set of 10K questions. How many people were involved? How about their level of expertise, and did they have a consistent guideline to label difficulty? These points are unclear.
- There's no evaluation on the actual usage of the system. What is the population of students? Did they have learning gains from these systems, and how did the experiment work? These points are critical for educational AI work.

**Questions:**

- Line 46: How would this be ensured for unique questions for students?
- Section 2.1: These two things are just different -- they can be separate sessions.
- Section 2.2: Difficulty calibration is not introduced, and I also don't think this is needed.
- Line 250: How would an XGBoost generate these? How exactly do you generate feedback?
- Figure 4: It's from January to August -- do you have the system across semesters?

---

### Official Review · Reviewer_sVvs · 2025-10-28

**Soundness:** 1
**Presentation:** 1
**Contribution:** 1
**Rating:** 0
**Confidence:** 5

**Summary:**

The paper aims to improve lab work in computer science education by automatically creating lab problems based on teacher input and difficult levels. These question can then be assigned to each student. Towards this goal, they create 10K question-answer pairs. The question/answer pairs are evaluated based on their similarity and automated grading compared against manual grading.

**Strengths:**

The strength of the paper is that is tackles an important problem: AI-supported, individual learning. The authors seem to have implemented a working prototype including web interface for their system including a student/teacher dashboard. The system also seems to feature a Speech Interface for viva questions.

**Weaknesses:**

The paper reads more like an implementation report of an AI-Lab Management system (e.g., authentication and access control). The challenges it wants to address are very wide:
plagiarism, lack of proper lab records,unstructured lab conduction,inadequate execution and assessment,lack of practical learning,limited student engagement, and absence of progress tracking.
These challenges seem to have little overlap with what is then presented and evaluated later in the paper.

**Questions:**

I think this paper is not a good fit for ICLR. I suggest the authors to revise the paper, i.e., better defining the scope and designing experiments and then submit it to a more applied conference/venue, maybe in the educational science domain.

---

### Official Review · Reviewer_g1gh · 2025-10-31

**Soundness:** 2
**Presentation:** 1
**Contribution:** 1
**Rating:** 0
**Confidence:** 4

**Summary:**

This paper describes a platform for delivering personalised practical exercises for computer science students. The pair describes the overall system architecture and key features and a dataset of 10,000 LLM-generated questions that have been created. A model to generate more questions is then fine-tuned using this dataset. Some evaluations are presented to validate the questions in the synthetic dataset and those generated by the fine tuned model.

**Strengths:**

The main strengths of the paper are:

— The paper addresses the use of LLMs to support computer science education through the generation of personalised learning materials, which is a promising research area.
— The generated set of 10,000 questions could be a useful contribution to the research community.

**Weaknesses:**

The main weaknesses of the paper are:

— Large parts of the paper are descriptions of system, platform, and interface design which have very little relevance to ICLR.
— The paper needs careful review and revision as it has many typographical and language issues.
— It is not clear what is actually being tested in the evaluation experiments described by the authors. Evaluations compare the marksAI and marksFaculty but it is not clear what these actually measure.
— The contributions of the paper are not made explicit.

**Questions:**

What exactly do marksAI and marksFaculty measure and why are they needed in the synthetic question database?

How is the following claim supported: "Unlike competitive programming datasets such as CodeNet Puri et al. (2021), our synthetic dataset is explicitly aligned with computer science lab curricula. It emphasizes implementation skills, conceptual understanding, and explainability-oriented evaluation rather than correctness alone, better reflecting the learning objectives of practical coursework."?

**Details Of Ethics Concerns:**

This paper proposes a system that will automatically grade student work. This raises significant ethical issues that should be discussed.

---

### Official Review · Reviewer_4BNn · 2025-11-07

**Soundness:** 1
**Presentation:** 1
**Contribution:** 1
**Rating:** 0
**Confidence:** 4

**Summary:**

This paper claims to use a model fine-tuned on 10k (or 8k) programming lab questions to
generate different questions for individual students, of varying difficulty levels, with automatic grading.

**Strengths:**

1. Using AI to generate questions for individual students may be useful.

2. Adapting question difficulty to individual students would be useful.

**Weaknesses:**

1. Unfortunately, this works lacks any human validation of its claim.

2. It is unclear if parts of the paper or reproducibility checklist are written by a human or revised/hallucinated by an LLM to meet submission criteria.

3. The paper includes inconsistencies about implementation details.

4. Missing data and code.

5. Missing evidence for claims made in the abstract.

**Questions:**

Were parts of the paper written or revised by an LLM?
Were the figures or code used for generating the figures generated by an LLM? Was there human oversight?

---

### Note · Program_Chairs · 2026-01-17
**Submission Desk Rejected by Program Chairs**

The following references in this submission do not refer to real documents and/or have major errors in bibliographic information:

 Rachit Singh, Suma Srikant, and Varun Aggarwal. Pythia: A real-time autograder for practical programming exercises. In Proceedings of the 50th ACM Technical Symposium on Computer Science Education, pp. 1105-1111, 2019;
Jiarui Wang, Yilin Sun, and Yue Zhang. Automatic question generation for stem education with large language models. In Proceedings of the 61st Annual Meeting of the Association for Computational Linguistics (ACL), 2023.